# Degradative Effect of Nattokinase on Spike Protein of SARS-CoV-2

**DOI:** 10.3390/molecules27175405

**Published:** 2022-08-24

**Authors:** Takashi Tanikawa, Yuka Kiba, James Yu, Kate Hsu, Shinder Chen, Ayako Ishii, Takami Yokogawa, Ryuichiro Suzuki, Yutaka Inoue, Masashi Kitamura

**Affiliations:** 1Laboratory of Nutri-Pharmacotherapeutics Management, School of Pharmacy, Faculty of Pharmacy and Pharmaceutical Sciences, Josai University, Saitama 350-0295, Japan; 2Laboratory of Pharmacognocy, School of Pharmacy, Faculty of Pharmacy and Pharmaceutical Sciences, Josai University, Saitama 350-0295, Japan; 3Contek Life Science Co., Ltd., Taipei City 100007, Taiwan; 4CellMark Japan, Tokyo 102-0071, Japan; 5Laboratory of Natural Products & Phytochemistry, Department of Pharmaceutical Sciences, Faculty of Pharmacy and Pharmaceutical Sciences, Josai University, Saitama 350-0295, Japan

**Keywords:** SARS-CoV-2, nattokinase, COVID-19

## Abstract

The coronavirus disease 2019 (COVID-19), caused by the severe acute respiratory syndrome coronavirus 2 (SARS-CoV-2), emerged as a pandemic and has inflicted enormous damage on the lives of the people and economy of many countries worldwide. However, therapeutic agents against SARS-CoV-2 remain unclear. SARS-CoV-2 has a spike protein (S protein), and cleavage of the S protein is essential for viral entry. Nattokinase is produced by *Bacillus subtilis* var. *natto* and is beneficial to human health. In this study, we examined the effect of nattokinase on the S protein of SARS-CoV-2. When cell lysates transfected with S protein were incubated with nattokinase, the S protein was degraded in a dose- and time-dependent manner. Immunofluorescence analysis showed that S protein on the cell surface was degraded when nattokinase was added to the culture medium. Thus, our findings suggest that nattokinase exhibits potential for the inhibition of SARS-CoV-2 infection via S protein degradation.

## 1. Introduction

Coronavirus disease 2019 (COVID-19), caused by severe acute respiratory syndrome coronavirus 2 (SARS-CoV-2), has been spreading worldwide. The COVID-19 pandemic has affected over 437 million people and caused more than 6.3 million deaths (https://covid19.who.int/, accessed on 4 July 2022). The entry of SARS-CoV-2 into host cells is mediated by the transmembrane spike protein (S protein), which forms homotrimers that extend from the viral envelope. The S protein is processed and activated by cellular proteases including transmembrane serine protein 2 (TMPRSS2), cathepsin, and furin. It comprises two functional subunits, S1 and S2; the S1 subunit of SARS-CoV-2 initiates virus-receptor binding by interacting with the human host cell receptor angiotensin-converting enzyme 2 (ACE2), and the S2 subunit participates in viral fusion with the target cell, allowing viral entry [1]. The receptor-binding domain (RBD) in the S1 subunit is responsible for binding to ACE2. S protein cleavage occurs at the boundary between the S1 and S2 subunits.

Currently, many countries are involved in the development of vaccines to protect against SARS-CoV-2 infection, thus, the number of SARS-CoV-2 infections has decreased. However, numerous variants of SARS-CoV-2, including strains with mutated vaccine target epitopes, have been reported [2,3]. Vaccination may not completely protect against SARS-CoV-2 infection because the number of patients with COVID-19 is increasing after vaccination. Therefore, it is important to develop novel treatments for SARS-CoV-2 infections.

Natto is a popular traditional Japanese food made from soybeans fermented by *Bacillus subtilis* var. *natto*. Nattokinase is found in natto [4] and is one of the most important extracellular enzymes produced by *B. subtilis* var. *natto* [5]. Nattokinase consists of 275 amino acids and is approximately 28 kDa [6,7]. Nattokinase inactivates plasminogen activator inhibitor-1 and increases fibrinolysis [8]. It also decreases the plasma levels of fibrinogen, factor VII, cytokines, and factor VIII [9]. Nattokinase has the highest clot-dissolving potency among naturally known anticoagulants [10]. A clinical trial demonstrated that oral consumption of nattokinase was not associated with any adverse effects [11]. Thus, nattokinase is now considered an efficient, secure, and economical enzyme that has drawn central attention in thrombolytic drug studies [12,13]. In addition, nattokinase is used in the treatment of some tumors [14,15].

A recent study revealed that natto extract inhibits bovine herpesvirus 1 (BHV-1) and SARS-CoV-2 infection [16]. These results indicate that natto extract protease might be effective against SARS-CoV-2 infection. In this study, we aimed to investigate whether the inhibition of SARS-CoV-2 infection by natto extract is caused by nattokinase derived from *B. subtilis* var. *natto*.

## 2. Results and Discussion

### 2.1. Degradative Effects of Nattokinase on Spike Protein of SARS-CoV-2 In Vitro

We first investigated whether nattokinase in natto extract could degrade SARS-CoV-2 S protein. The S protein of SARS-CoV-2 plays an important role in the ACE2 receptor of the host cell during the early stages of infection [17]. After mixing the S protein expression cell lysate with a 4-fold dilution series of nattokinase (32 µg/mL, 8 µg/mL, 2 µg/mL, 500 ng/mL, 125 ng/mL, 31.25 ng/mL, and 7.8125 ng/mL), Western blotting was performed. The full length of S protein (S1 and S2 subunits) and S2 subunit appeared as bands when S protein expression cell lysate was incubated with D-PBS at nattokinase concentrations of 500 ng/mL, 125 ng/mL, 31.25 ng/mL, and 7.8125 ng/mL (Figure 1A). Next, we examined whether nattokinase degrades the S protein in a time-dependent manner. The lysate was then incubated with 1 µg/mL nattokinase for 10–180 min. The S protein of SARS-CoV-2 was degraded by nattokinase after 60–180 min of incubation, but not after 10 and 30 min of incubation (Figure 1B). Thus, nattokinase degraded S protein in a dose- and time-dependent manner.

To confirm whether the degradative effect of nattokinase is due to enzymatic activity, nattokinase was treated with heating or a protease inhibitor cocktail. When nattokinase was heated at 100 °C for 5 min, the degradative effect of nattokinase was lost (Figure 1C, lane 6). Furthermore, the loss of the S protein bands by nattokinase was blocked when protease inhibitors were added (Figure 1C, lanes 4 and 5). Compared with protein inhibitor cocktail I, protein cocktail III, which consisted of AEBSF HCl (4-(2-Aminoethyl) benzenesulfonyl fluoride hydrochloride), aprotinin, which is an irreversible serine protease inhibitor, and leupeptin, which is a cysteine-protease, clearly blocked nattokinase activity. Nattokinase has the same conserved amino acids, Ser-His-Asp (Asp^32^, His^64^, and Ser^221^), which are members of the subtilisin family of serine proteases [6,18]. The crystal structure of nattokinase is nearly identical to that of subtilisin E from *B. subtilis* DB104 [19]. This result is consistent with that of a previous report that nattokinase is a serine protease. We also assessed the degradative effects of nattokinase using cell lysates expressing the RBD and ACE2. When 7.5 µg/mL of nattokinase and cell lysate were incubated, the bands of RBD and ACE2 were lost (Figure 1D).

### 2.2. Degradative Effects of Nattokinase on Spike Protein of SARS-CoV-2 on the Transfected Cell Surface

Next, we examined whether nattokinase degrades the S protein on the transfected cell surface. The S protein was transfected with the HEK293 cells. The transfected cells were incubated with nattokinase for 9 h. The S protein on the cell surface was detected using an anti-S protein antibody without cell permeabilization (Figure 2A). The S protein was detected in the transfected cells. When transfected cells were treated with nattokinase, the S protein on the cell surface decreased. When cells were treated with 25 µg/mL and 2.5 µg/mL nattokinase, the ratio of S protein-positive area to nucleus-positive area decreased by approximately 0.3 and 0.7, respectively (Figure 2B). The degradative effect of nattokinase was observed when there was no cytotoxicity (Figure 2C). Western blotting analysis showed that the quantity of total S protein did not change among nattokinase and control treatments (Appendix A). These results indicate that nattokinase would degrade the S protein of SARS-CoV-2 in the non-toxic concentration range.

In this study, we showed that the protease activity of nattokinase contributes to the degradation of S protein. Nattokinase has a degrading effect on not only S proteins but also ACE2 in host cells. The protease specificity of nattokinase would be low, because GAPDH, a housekeeping protein, was also degraded simultaneously in the in-vitro evaluation of nattokinase mixed with cell lysate (Appendix A). On the other hand, when added to cells, it does not show any effect on cell viability and is expected to act as a protective agent on the cell surface. Further analysis of the degradation products of nattokinase using mass spectrometry is needed for understanding the proteolysis effects.

Nattokinase possesses the potent degradation activity for SARS-CoV-2 S protein and has also been shown to exert anti-atherosclerotic, lipid-lowering, antihypertensive, antithrombotic, fibrinolytic, neuroprotective, antiplatelet, and anticoagulant effects [20]. Patients with hypertension and cardiovascular comorbidities can easily get very sick from COVID-19 [21]. Due to the emergence of numerous variants of SARS-CoV-2 including strains with mutated vaccine target epitopes, vaccination alone may not completely protect against SARS-CoV-2 infection. Nattokinase and natto extracts have the potential to be developed as a new generation of drug for the prevention and treatment of COVID-19.

## 3. Materials and Methods

### 3.1. Materials

Nattokinase was obtained from Contek Life Science Co., Ltd. (Taipei City, Taiwan). The nattokinase activity was 60,000 FU/g (FU, fibrinolysis unit). Protease inhibitor cocktail sets I and III were purchased from FUJIFILM Wako Pure Chemical Corporation (Osaka, Japan). The expression plasmid (pcDNA3.1-SARS2-Spike C9 with tag at the C-terminal, pcDNA3.1-hACE2, and pcDNA3-SARS-CoV-2-S-RBD-sfGFP) was purchased from Addgene (Watertown, MA, USA). HEK293 (JCRB9068) cells were obtained from the JCRB Cell Bank (Osaka, Japan).

### 3.2. Cell Culture and Western Blotting

HEK293 cells were cultured at a density of 3.5 × 10^5^ cells/mL in DMEM supplemented with 10% FBS, L-glutamine, 100 U/mL penicillin, and 100 μg/mL streptomycin overnight. The cells were transfected with each plasmid (pcDNA3.1-SARS2-Spike, pcDNA3-SARS-CoV-2-S-RBD-sfGFP, or pcDNA3.1-hACE2) and incubated for 22 h. After incubation, the cultured cells were scraped and washed with ice-cold Dulbecco’s phosphate-buffered saline (D-PBS). Cell counting was performed and xTractor buffer (Takara Bio Inc., Shiga, Japan) was added to the cell precipitate. The cell lysates were centrifuged at 1300× *g* for 10 min at 4 °C and the supernatant was transferred to new tubes and stored at −80 °C until use. The protein concentration was determined by bicinchoninic acid (BCA) protein assay using a BCA assay kit (Takara). Ten microliters of nattokinase and 10 µL of cell lysate (1 μ µg/mL) were incubated at 37 °C for 1 h. When the effects of protease inhibitors were used, protease inhibitor cocktail sets I and III were diluted 10-fold with D-PBS, and a 10 µL protease inhibitor cocktail solution was added to the mixture of nattokinase and cell lysate. Equal volumes of the reaction mixture were loaded and Western blotting was performed. The primary antibodies included anti-rhodopsin (C9) mouse monoclonal antibody (1D4) (Santa Cruz Biotechnology, Dallas, TX, USA), anti-GAPDH mouse monoclonal antibody (FUJIFILM Wako), anti-GFP tag mouse monoclonal antibody (Proteintech, Rosemont, IL, USA), and anti-ACE2 antibody (Proteintech). Secondary antibodies include HRP-conjugated goat anti-mouse antibody (Proteintech).

### 3.3. Immunofluorescence Assay

HEK293 cells were cultured at a density of 3.5 × 10^5^ cells/mL in an 8-well chamber in DMEM supplemented with 10% FBS, L-glutamine, 100 U/mL penicillin, and 100 μg/mL streptomycin. The cells were transfected with pcDNA3.1-SARS2-Spike and incubated for 9 h. After incubation, the cells were treated with the samples, incubated for 13 h, and fixed in 4% paraformaldehyde for 30 min. After incubation with SARS-CoV/SARS-CoV-2 spike monoclonal antibody (1A9) (GeneTex, CA, USA) for 1 h, it was incubated with Cy3–conjugated goat anti-mouse antibody for 1 h. The slides were stained with DAPI Fluoromount-G and observed using a fluorescence microscope (BZ-X710, Keyence, Osaka, Japan). S protein-positive and nucleus-positive areas were calculated using BZ-X710 attached analysis software (BZ-X Analyzer). Cell viability was assessed using the 3-(4,5-Dimethylthiazol-2-yl)-2,5-diphenyltetrazolium bromide (MTT) assay. The cells were cultivated in 24-well culture plates. After incubation at 37 °C for 24 h, samples were added to each well and incubated for another 13 h. Cells were suspended in 500 μL of DMEM containing 500 μg/mL MTT. After incubation for 3 h at 37 °C, 500 μL isopropanol containing 4 mM HCl was added to dissolve MTT formazan. The absorbance was measured at 570 nm using a microplate reader.

## 4. Conclusions

In this study, we demonstrated that nattokinase, a serine protease, degrades the S protein of SARS-CoV-2. To investigate whether nattokinase contained in natto extract could inhibit SARS-CoV-2 infection, we analyzed S protein degradation by mixing the S protein expression cell lysate and nattokinase in a dose- and time-dependent manner. The RBD of the S protein binds to the membrane-distal portion of the ACE2 protein. Natto extract has been reported to inhibit SARS-CoV-2 infection in Vero E6 cells via RBD degradation [16]. We demonstrated that S protein degradation by nattokinase was blocked by heat or protein-inhibitor treatments. Our data suggest that the protease activity of nattokinase plays a crucial role in S protein degradation. Taken together, these findings support the notion that the inhibition of SARS-CoV-2 infection by natto extract was due to S protein degradation by nattokinase. Thus, our data indicated that nattokinase and natto extracts have potential effects on the inhibition of SARS-CoV-2 host cell entry via S protein degradation.

## Figures and Tables

**Figure 1 molecules-27-05405-f001:**
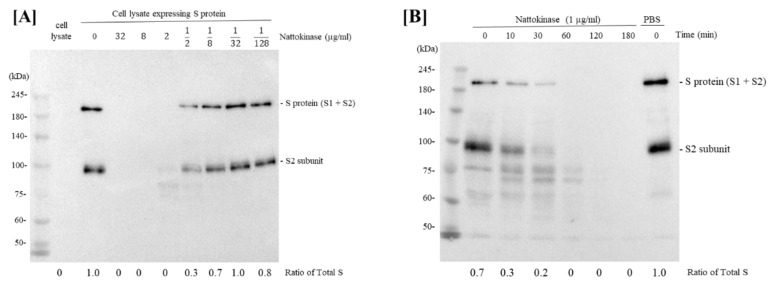
(**A**) Degradative effects of nattokinase in dose-dependent manner. Serial diluted nattokinase (32 µg/mL, 8 µg/mL, 2 µg/mL, 500 ng/mL, 125 ng/mL, 31.25 ng/mL, and 7.8125 ng/mL) were mixed with S protein expression cell lysate and incubated. Full length of S protein (S1 and S2 subunits) and S2 subunit were detected as upper and lower bands, respectively. Ratio of total S was indicated as the relative quantity of S protein (S protein + S2 protein). (**B**) Degradative effects of nattokinase in time-dependent manner. S protein expression cell lysate was incubated with 1 µg/mL nattokinase for 0, 10, 30, 60, 120, and 180 min. (**C**) Effects of heating treatment or protease inhibitors. Lane 1: HEK293 lysate; lane 2: HEK293 lysate (S protein); lane 3: HEK293 (S protein) + nattokinase (5 µg/mL); lane 4: HEK293 (S protein) + nattokinase (5 µg/mL) + Protease inhibitor I; lane 5: HEK293 (S protein) + nattokinase (5 µg/mL) + Protease inhibitor III; lane 6: HEK293 (S protein) + heat-treated nattokinase (5 µg/mL). (**D**) Degradative effect on RBD of S protein and ACE2. RBD of S protein and ACE2 coding plasmids were transfected with HEK293 cells, respectively. Cell lysates were incubated with nattokinase (7.5 µg/mL) and heat-treated nattokinase (7.5 µg/mL) and Western blotting was performed.

**Figure 2 molecules-27-05405-f002:**
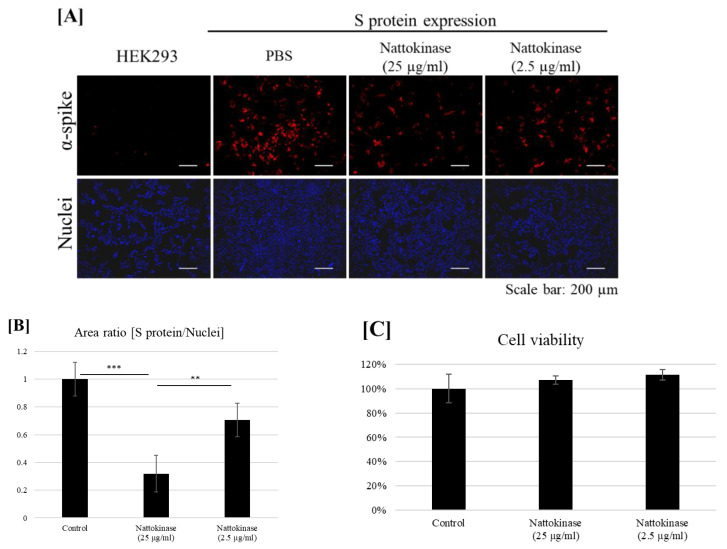
(**A**) Degradative effect of nattokinase on S protein on the cell surface. Spike-pcDNA3.1 was transfected with HEK293 cells and incubated for 9 h. After incubation, nattokinase (25 and 2.5 µg/mL) were added to culture medium and further incubated for 13 h. Cells were fixed and immunofluorescent analysis was performed. S protein on the cell surface was stained with anti-spike protein antibody (Red) and nucleus was stained with DAPI (Blue). (**B**) Ratio of S protein area to nucleus positive area. Three images per sample were captured and S protein/nucleus positive areas were calculated. Data are shown as mean + SD, and *p*-value was determined by one-way analysis of variance (ANOVA) with Tukey’s post-hoc test using R software (R-3.3.3 for windows) (** *p* < 0.01; *** *p* < 0.001). (**C**) Cell viability was evaluated by MTT assay. Indicated nattokinase was added to culture medium and incubated for 13 h; MTT assay was performed.

## Data Availability

The data used to support these findings have been included in this article. Additional information is available from the corresponding authors upon request.

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
