# Peer review of "Degradative Effect of Nattokinase on Spike Protein of SARS-CoV-2"

_molecules, 2022, doi:10.3390/molecules27175405_

Round 1

Reviewer 1 Report

Comments:

   In this manuscript, the authors described “Degradative effect of nattokinase on Spike protein of SARS-CoV-2”. This paper show that nattokinase exhibits potential for the inhibition of SARS-CoV-2 infection via S protein degradation. However, there are a few points that need to be clarified. 

Comment

1. Natto is well known to be nutritious and beneficial for health in Japan. Does Natto extract have degradation effect on SARS-CoV-2 spike protein? The author shall be justifying it.

2. In Figure 1, the author should quantify it.

3. In Figure 1C, the author should mark it clearly. Easy for readers to read.

4. In Figure 1D, what is NK?  The author should mark it clearly. 

5. The part of the discussion should be strengthened.

6. The format of the article written by the author is different from this journal molecule.

Author Response

We corrected the maniscript accroding to reviewer's comments.

Point-by-point response is described at the last page 10-11.

Reviewer 2 Report

The work of Takashi and co-authors is devoted to an important topic, namely the expansion of the set of methods to counter coronaviruses. Indeed, despite significant efforts to develop and introduce vaccines against covid-19, the ongoing evolution of the virus calls into question the effectiveness of using vaccines alone.

The authors suggest using serine proetase as an alternative therapeutic approach.

Despite the importance and necessity of this kind of research, I have a number of comments.

1. The aim of the work (we aimed to investigate whether the inhibition of SARS-CoV-2 infection by natto extract is caused by nattokinase derived from B. subtilis var. natto.) is not formulated correctly. There were no works with the virus in the article, therefore, it is impossible to draw conclusions about antiviral activity.

2. Obviously, nattokinase is not a specific protease, so using only S protein or RBD as a target is a manipulation. A heterologous protein control should be added to show non-specific proteolysis.

3. The quality of illustrations is low. First of all, you need to add molecular weight markers for each gel photo.

4. The statement in the conclusions (These results strongly suggest that nattokinase has the potential to degrade various types of mutations of SARS-CoV-2 that have emerged and could emerge in the future) is unfounded and does not follow from the results of the work.

My recommendation is to revise the article and remove the wording related to anti-virus activity. It would be nice to study proteolysis products using  mass spectrometry.

Author Response

Please see the attachment."

We corrected the maniscript accroding to reviewer's comments.

Point-by-point response is described at the last page 10-12.

Round 2

Reviewer 1 Report

accepted

Reviewer 2 Report

All comments have been corrected. The article can be recommended for publication.